# Differentially Private Bagging: Improved utility and cheaper privacy than subsample-and-aggregate

**James Jordon**
University of Oxford
james.jordon@wolfson.ox.ac.uk

**Jinsung Yoon**
University of California, Los Angeles
jsyoon0823@g.ucla.edu

**Mihaela van der Schaar**
University of Cambridge
University of California, Los Angeles
Alan Turing Institute
mv472@cam.ac.uk, mihaela@ee.ucla.edu

## Abstract

Differential Privacy is a popular and well-studied notion of privacy. In the era of big data that we are in, privacy concerns are becoming ever more prevalent and thus differential privacy is being turned to as one such solution. A popular method for ensuring differential privacy of a classifier is known as subsample-and-aggregate, in which the dataset is divided into distinct chunks and a model is learned on each chunk, after which it is aggregated. This approach allows for easy analysis of the model on the data and thus differential privacy can be easily applied. In this paper, we extend this approach by dividing the data several times (rather than just once) and learning models on each chunk within each division. The first benefit of this approach is the natural improvement of utility by aggregating models trained on a more diverse range of subsets of the data (as demonstrated by the well-known bagging technique). The second benefit is that, through analysis that we provide in the paper, we can derive tighter differential privacy guarantees when several queries are made to this mechanism. In order to derive these guarantees, we introduce the upwards and downwards moments accountants and derive bounds for these moments accountants in a data-driven fashion. We demonstrate the improvements our model makes over standard subsample-and-aggregate in two datasets (Heart Failure (private) and UCI Adult (public)).

## 1 Introduction

In the era of big data that we live in today, privacy concerns are becoming ever more prevalent. It falls to the researchers using the data to ensure that adequate measures are taken to ensure any results that are put into the public domain (such as the parameters of a model learned on the data) do not disclose sensitive attributes of the real data. For example, it is well known that the high capacity of deep neural networks can cause the networks to "memorize" training data; if such a network's parameters were made public, it may be possible to deduce some of the training data that was used to train the model, thus resulting in real data being leaked to the public.

Several attempts have been made at rigorously defining what it means for an algorithm, or an algorithm's output, to be "private". One particularly attractive and well-researched notion is that of differential privacy [1]. Differential privacy is a formal definition that requires that the distribution of the output of a (necessarily probabilistic) algorithm not be too different when a single data point is included in the dataset or not. Typical methods for enforcing differential privacy involve bounding

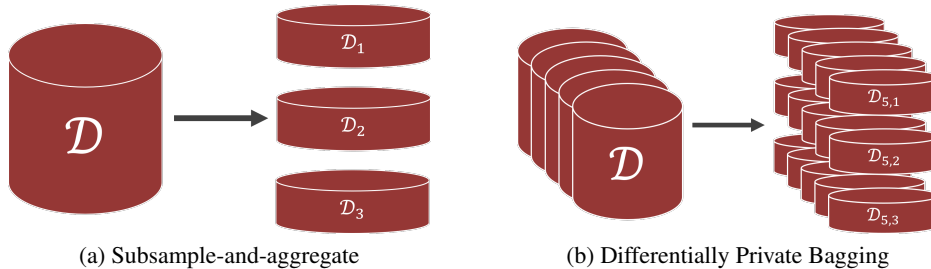

(a) Subsample-and-aggregate      (b) Differentially Private Bagging

Figure 1: A comparison of how the dataset is used in (a) subsample-and-aggregate and (b) our differentially private bagging procedure. By partitioning the dataset *multiple times* we are able to perform a tighter privacy analysis using our personalised moments accountant in addition to learning a better performing underlying classifier.

the effect that inclusion of a single sample can have on the output and then adding noise (typically Laplacian or Gaussian) proportional to this effect. The most difficult step in this process is in attaining a good bound on the effect of inclusion.

One method for bypassing this difficulty, is to build a classifier by dividing up the dataset into distinct subsets, training a separate classifier on each chunk, and then aggregating these classifiers. The effect of a single sample is then bounded by the fact that it was used only to train exactly one of these models and thus its inclusion or exclusion will affect only that model's output. By dividing the data into smaller chunks, we learn more models and thus the one model that a sample can effect becomes a smaller "fraction" of the overall model, thus resulting in a smaller effect that any one sample has on the model as a whole. This method is commonly referred to as subsample-and-aggregate [2, 3, 4].

In this work, we propose an extension to the subsample-and-aggregate methodology that has similarities with bagging [5]. Fig. 1 depicts the key methodological difference between standard subsample-and-aggregate and our proposed framework, Differentially Private Bagging (DPBag), namely that we partition the dataset many times. This multiple-partitioning not only improves utility by building a better predictor, but also enjoys stronger privacy guarantees due to the fact that the effect of adding or removing a single sample can be more tightly bounded within our framework. In order to prove these guarantees, we introduce the *personalised moments accountants*, which are data-driven variants of the moments accountant [6], that allow us to track the privacy loss with respect to each sample in the dataset and then deduce the final privacy loss by taking the maximum loss over all samples. The personalised moments accountant also lends itself to allowing for personalised differential privacy [7] in which we may wish to allow each individual to specify their own privacy parameters.

We demonstrate the efficacy of our model on two classification tasks, demonstrating that our model is an improvement over the standard subsample-and-aggregate algorithm.

## 2 Related Works

Several works have proposed methods for differentially private classification. Of particular interest is the method of [6], in which they propose a method for differentially private training of deep neural networks. In particular, they introduce a new piece of mathematical machinery, the moments accountant. The moments accountant allows for more efficient composition of differentially private mechanisms than either simple or advanced composition [1]. Fortunately, the moments accountant is not exclusive to deep networks and has proven to be useful in other works. In this paper, we use two variants of the moments accountant, which we refer to collectively as the personalised moments accountants. Our algorithm lends itself naturally to being able to derive tighter bounds on these personalised moments accountants than would be possible on the "global" moments accountant.

Most other methods use the subsample-and-aggregate framework (first discussed in [2]) to guarantee differential privacy. A popular, recent subsample-and-aggregate method is Private Aggregation of Teacher Ensembles (PATE), proposed in [8]. Their main contribution is to provide a data-driven bound on the moments accountant for a given query to the subsample-and-aggregate mechanism that

they claim significantly reduces the privacy cost over the standard data-independent bound. This is further built on in [9] by adding a mechanism that first determines whether or not a query will be too expensive to answer or not, only answering those that are sufficiently cheap. Both works use standard subsample-and-aggregate in which the data is partitioned only once. Our method is more fundamental than PATE, in the sense that the techniques used by PATE to improve on subsample-and-aggregate would also be applicable to our differentially private bagging algorithm. The bound they derive in [8] on the moments accountant should translate to our personalised moments accountants in the same way the data-independent bound does (i.e. by multiplying the dependence on the inverse noise scale by a data-driven value) and as such our method would provide privacy improvements over PATE similar to the improvements it provides over standard subsample-and-aggregate. We give an example of our conjectured result for PATE in the Supplementary Materials for clarity.

Another method that utilises subsample-and-aggregate is [10], in which they use the distance to instability framework [4] combined with subsample-and-aggregate to privately determine whether a query can be answered *without* adding any noise to it. In cases where the query can be answered, no privacy cost is incurred. Whenever the query cannot be answered, no answer is given but a privacy cost is incurred. Unfortunately, the gains to be had by applying our method over basic subsample-and-aggregate to their work are not clear, but we believe that at the very least, the utility of the answer provided may be improved on due to the ensemble having a higher utility in our case (and the same privacy guarantees will hold that they prove).

In [11], they build a method for learning a differentially private decision tree. Although they apply bagging to their framework, they do not do so to create privacy, but only to improve the utility of their learned classifier. The privacy analysis they provide is performed only on each individual tree and not on the ensemble as a whole.

# 3 Differential Privacy

Let us denote the feature space by $\mathcal{X}$, the set of possible class labels by $\mathcal{C}$ and write $\mathcal{U} = \mathcal{X} \times \mathcal{C}$. Let us denote by $D$ the collection of all possible datasets consisting of points in $\mathcal{U}$. We will write $\mathcal{D}$ to denote a dataset in $D$, so that $\mathcal{D} = \{u_i\}_{i=1}^N = \{(x_i, y_i)\}_{i=1}^N$ for some $N$.

We first provide some preliminaries on differential privacy [1] before describing our method; we refer interested readers to [1] for a thorough exposition of differential privacy. We will denote an algorithm by $\mathcal{M}$, which takes as input a dataset $\mathcal{D}$ and outputs a value from some output space, $\mathcal{R}$.

**Definition 1** (Neighboring Datasets [1]). Two datasets $\mathcal{D}, \mathcal{D}'$ are said to be neighboring if

$$\exists u \in \mathcal{U} \text{ s.t. } \mathcal{D} \setminus \{u\} = \mathcal{D}' \text{ or } \mathcal{D}' \setminus \{u\} = \mathcal{D}.$$

**Definition 2** (Differential Privacy [1]). A *randomized* algorithm, $\mathcal{M}$, is $(\epsilon, \delta)$-differentially private if for all $\mathcal{S} \subset \mathcal{R}$ and for all neighboring datasets $\mathcal{D}, \mathcal{D}'$:

$$\mathbb{P}(\mathcal{M}(\mathcal{D}) \in \mathcal{S}) \leq e^\epsilon \mathbb{P}(\mathcal{M}(\mathcal{D}') \in \mathcal{S}) + \delta$$

where $\mathbb{P}$ is taken with respect to the randomness of $\mathcal{M}$.

Differential privacy provides an intuitively understandable notion of privacy - a particular sample's inclusion or exclusion in the dataset does not change the probability of a particular outcome very much: it does so by a multiplicative factor of $e^\epsilon$ and an additive amount, $\delta$.

# 4 Differentially Private Bagging

In order to enforce differential privacy, we must bound the effect of a sample's inclusion or exclusion on the output of the model. In order to do this, we propose a model for which the maximal effect can be easily deduced and moreover, for which we can actually show a lesser maximal effect by analysing the training procedure and deriving data-driven privacy guarantees.

We begin by considering $k$ (random) partitions of the dataset, $\mathcal{D}_1, ..., \mathcal{D}_k$ with $\mathcal{D}_i = \{D_1^i, ..., D_n^i\}$ for each $i$, where $D_j^i$ is a set of size $\lfloor \frac{|\mathcal{D}|}{n} \rfloor$ or $\lceil \frac{|\mathcal{D}|}{n} \rceil$. We then train a "teacher" model, $T_{ij}$ on each of these sets (i.e. $T_{ij}$ is trained on $D_j^i$). We note that each sample $u \in \mathcal{D}$ is in precisely one set from each partition and thus in precisely $k$ sets overall; it is therefore used to train $k$ teachers. We

collect the indices of the corresponding teachers in the set $I(u) = \{(i,j) : u \in D^i_j\}$ and denote by $\mathcal{T}(u) = \{T_{ij} : (i,j) \in I(u)\}$ the set of teachers trained using the sample $u$.

Given a new sample to classify $x \in \mathcal{X}$, we first compute for each class the number of teachers that output that class, $n_c(x) = |\{(i,j) : T_{ij}(x) = c\}|$. The model then classifies the sample as

$$\hat{c}(x) = \arg\max\{n_c(x) : c \in \mathcal{C}\}$$

i.e. by classifying it as the class with the most votes. To make the output differentially private, we can add independent Laplacian noise to each of the resulting counts before taking $\arg\max$. So that the classification becomes

$$\tilde{c}_\lambda(x) = \arg\max\{n_c(x) + Y_c : c \in \mathcal{C}\}$$

where $Y_c, c \in \mathcal{C}$ are independent $Lap(\frac{k}{\lambda})$ random variables and where $\lambda$ is a hyper-parameter of our model. We scale the noise to the number of partitions because the number of partitions is precisely the total number of teachers that any individual sample can effect. Thus the (naive) bound on the $\ell_1$-sensitivity of this algorithm is $k$, giving us the following theorem, which tells us that our differentially private bagging algorithm is *at least* as private as the standard subsample-and-aggregate mechanism, independent of the number of partitions used.

**Theorem 1.** With $k$ partitions and $n$ teachers per partition, $\tilde{c}_\lambda$ is $2\lambda$-differentially private with respect to the data $\mathcal{D}$.

*Proof.* This follows immediately from noting that the $\ell_1$-sensitivity of $n_c(x)$ is $k$. See [1]. $\square$

We note that the standard subsample-and-aggregate algorithm can be recovered from ours by setting $k = 1$. In the next section, we will derive tighter bounds on the differential privacy of our bagging algorithm when several queries are made to the classifier.

## 4.1 Personalised Moments Accountants

In order to provide tighter differential privacy guarantees for our method, we now introduce the personalised moments accountants. Like the original moments accountant from [6], these will allow us to compose a sequence of differentially private mechanisms more efficiently than using standard or advanced composition [1]. We begin with a preliminary definition (found in [6]).

**Definition 3** (Privacy Loss and Privacy Loss Random Variable [6]). Let $\mathcal{M} : D \to \mathcal{R}$ be a randomized algorithm, with $\mathcal{D}$ and $\mathcal{D}'$ a pair of neighbouring datasets. Let $aux$ be any auxiliary input. For any outcome $o \in \mathcal{R}$, we define the privacy loss at $o$ to be:

$$c(o; \mathcal{M}, aux, \mathcal{D}, \mathcal{D}') = \log \frac{\mathbb{P}(\mathcal{M}(\mathcal{D}, aux) = o)}{\mathbb{P}(\mathcal{M}(\mathcal{D}', aux) = o)}$$

with the privacy loss *random variable*, $C$, being defined by

$$C(\mathcal{M}, aux, \mathcal{D}, \mathcal{D}') = c(\mathcal{M}(\mathcal{D}, aux), aux, \mathcal{D}, \mathcal{D}')$$

i.e. the random variable defined by evaluating the privacy loss at a sample from $\mathcal{M}(\mathcal{D}, aux)$.

In defining the moments accountant, an intermediate quantity, referred to by [6] as the "$l$-th moment" is introduced. We divide the definition of this $l$-th moment into a downwards and an upwards version (corresponding to whether $\mathcal{D}'$ is obtained by either removing or adding an element to $\mathcal{D}$, respectively). We do this because the upwards moments accountant must be bounded among *all* possible points $u \in \mathcal{U}$ that could be added, whereas the downwards moments accountants need only consider the points that are already in $\mathcal{D}$.

**Definition 4.** Let $\mathcal{D}$ be some dataset and let $u \in \mathcal{D}$. Let $aux$ be any auxiliary input. Then the downwards moments accountant is given by

$$\check{\alpha}_{\mathcal{M}}(l; aux, \mathcal{D}, u) = \log \mathbb{E}(\exp(lC(\mathcal{M}, aux, \mathcal{D}, \mathcal{D} \setminus \{u\}))).$$

**Definition 5.** Let $\mathcal{D}$ be some dataset. Then the upwards moments accountant is defined as

$$\hat{\alpha}_{\mathcal{M}}(l; aux, \mathcal{D}) = \max_{u \in \mathcal{U}} \log \mathbb{E}(\exp(lC(\mathcal{M}, aux, \mathcal{D}, \mathcal{D} \cup \{u\}))).$$

We can recover the original moments accountant from [6], $\alpha_{\mathcal{M}}(l)$, as

$$\alpha_{\mathcal{M}}(l) = \max_{aux, \mathcal{D}}\{\hat{\alpha}_{\mathcal{M}}(l; aux, \mathcal{D}), \max_u \check{\alpha}_{\mathcal{M}}(l; aux, \mathcal{D}, u)\}. \tag{1}$$

We will use this fact, together with the two theorems in the following subsection, to calculate the final global privacy loss of our mechanism.

## 4.2 Results inherited from the Moments Accountant

The following two theorems state two properties that our personalised moments accountants share with the original moments accountant. Note that the composability in Theorem 2 is being applied to *each* personalised moments accountant individually.

**Theorem 2** (Composability). Suppose that an algorithm $\mathcal{M}$ consists of a sequence of adaptive algorithms (i.e. algorithms that take as auxiliary input the outputs of the previous algorithms) $\mathcal{M}_1, ..., \mathcal{M}_m$ where $\mathcal{M}_i : \prod_{j=1}^{i-1} \mathcal{R}_j \times D \to \mathcal{R}_i$. Then, for any $l$

$$\check{\alpha}_{\mathcal{M}}(l; \mathcal{D}, u) \leq \sum_{i=1}^{m} \check{\alpha}_{\mathcal{M}_i}(l; \mathcal{D}, u)$$

and

$$\hat{\alpha}_{\mathcal{M}}(l; \mathcal{D}) \leq \sum_{i=1}^{m} \hat{\alpha}_{\mathcal{M}_i}(l; \mathcal{D}).$$

*Proof.* The statement of this theorem is a variation on Theorem 2 from [6], applied to the personalised moments accountants. Their proof involves proving this stronger result. See [6], Theorem 2 proof. □

**Theorem 3** (($\epsilon, \delta$) from $\alpha(l)$ [6]). Let $\delta > 0$. Any mechanism $\mathcal{M}$ is ($\epsilon, \delta$)-differentially private for

$$\epsilon = \min_l \frac{\alpha_{\mathcal{M}}(l) + \log(\frac{1}{\delta})}{l} \tag{2}$$

*Proof.* See [6], Theorem 2. □

Theorem 2 means that bounding each personalised moments accountant individually could provide a significant improvement on the overall bound for the moments accountant. Combined with Eq. 1, we can first sum over successive steps of the algorithm *and then* take the maximum. In contrast, original approaches that bound only the overall moments accountant at each step essentially compute

$$\alpha_{\mathcal{M}}(l) = \sum_{i=1}^{m} \max_{aux, \mathcal{D}} \{\hat{\alpha}_{\mathcal{M}_i}(l; aux, \mathcal{D}), \max_u \check{\alpha}_{\mathcal{M}_i}(l; aux, \mathcal{D}, u)\}. \tag{3}$$

Our approach of bounding the personalised moments accountant allows us to compute the bound as

$$\alpha_{\mathcal{M}}(l) = \max_{aux, \mathcal{D}} \{\sum_{i=1}^{m} \hat{\alpha}_{\mathcal{M}_i}(l; aux, \mathcal{D}), \max_u \sum_{i=1}^{m} \check{\alpha}_{\mathcal{M}_i}(l; aux, \mathcal{D}, u)\} \tag{4}$$

which is strictly smaller whenever there is not some personalised moments accountant that is *always* larger than *all* other personalised moments accountants. The bounds we derive in the following subsection and the subsequent remarks will make clear why this is an unlikely scenario.

## 4.3 Bounding the Personalised Moments Accountants

Having defined the personalised moments accountants, we can now state our main theorems, which provide a data-dependent bound on the personalised moments accountant for a single query to $\tilde{c}_\lambda$.

**Theorem 4** (Downwards bound). Let $x_{new} \in \mathcal{X}$ be a new point to classify. For each $c \in \mathcal{C}$ and each $u \in \mathcal{D}$, define the quantities

$$n_c(x_{new}; u) = \frac{|\{(i, j) \in I(u) : T_{ij}(x_{new}) = c\}|}{k}$$

i.e. $n_c(x_{new}; u)$ is the fraction of teachers *that were trained on a dataset containing $u$* that output class $c$ when classifying $x_{new}$. Let

$$m(x_{new}; u) = \max_c \{1 - n_c(x_{new}; u)\}.$$

Then

$$\check{\alpha}_{\tilde{c}_\lambda(x_{new})}(l; \mathcal{D}, u) \leq 2\lambda^2 m(x_{new}; u)^2 l(l+1). \tag{5}$$

*Proof.* (Sketch.) The theorem follows from the fact that $m(x_{new}; u)$ is the maximum change that can occur in the vote fractions, $n_c, c \in \mathcal{C}$ when the sample $u$ is removed from the training of each model in $\mathcal{T}(u)$, corresponding to all teachers that were not already voting for the minority class switching their vote to the minority class. $m$ can thus be thought of as the personalised $\ell_1$-sensitivity of a specific query to our algorithm, and so the standard sensitivity based argument gives us that $\tilde{c}_\lambda(x_{new})$ is $2\lambda m(x_{new}; u)$-differentially private *with respect to removing* $u$. The bound on the (downwards) moments accountant then follows using a similar argument to the proof of Prop. 3.3 in [12]. $\square$

To prove the upwards bound, we must understand what happens when we add a point to our training data - which is that it will be added to a training set for precisely 1 teacher in each of the $k$ partitions. Each dataset in a partition will either be of size $\lceil \frac{|\mathcal{D}|}{n} \rceil$ or $\lfloor \frac{|\mathcal{D}|}{n} \rfloor$. We assume (without loss of generality) that a new point is added to the first dataset in each partition that contains $\lfloor \frac{|\mathcal{D}|}{n} \rfloor$ samples. We collect the indices of these datasets in $I(*)$ and denote the set of teachers trained on these subsets by $\mathcal{T}(*)$.

**Theorem 5** (Upwards bound). Let $x_{new} \in \mathcal{X}$ be a new point to classify. For each $c \in \mathcal{C}$, define the quantity

$$n_c(x_{new}; *) = \frac{|\{(i,j) \in I(*) : T_{ij}(x_{new}) = c\}|}{k}$$

i.e. $n_c(x_{new}; *)$ is the fraction of teachers *whose training set would receive the new point* that output class $c$ when classifying $x_{new}$. Let

$$m(x_{new}; *) = \max_c \{1 - n_c(x_{new}; *)\}.$$

Then

$$\hat{\alpha}_{\tilde{c}_\lambda(x_{new})}(l; \mathcal{D}) \leq 2\lambda^2 m(x_{new}; *)^2 l(l+1). \tag{6}$$

*Proof.* The proof is exactly as for Theorem 4, replacing $I(u)$ and $\mathcal{T}(u)$ with $I(*)$ and $\mathcal{T}(*)$. $\square$

The standard bound on the moments accountant of a $2\lambda$ differentially private algorithm is $2\lambda^2 l(l+1)$ (see [12]). Thus, our theorems introduce a factor of $m(x_{new}; u)^2$. Note that by definition $m \leq 1$ and thus our bound is in general tighter, but always at least as tight. It should be noted, however, that for a single query, this bound may not improve on the naive $2\lambda^2 l(l+1)$ bound, since in that case equations 3 and 4 are equal. If there is *any* training sample $u \in \mathcal{D} \cup \{*\}$ and any class $c \in \mathcal{C}$ for which *all* teachers in $\mathcal{T}(u)$ classify $x_{new}$ as some class other than $c$ then $m(x_{new}; u) = 1$. However, over the course of several queries, it is unlikely that each set of teachers $\mathcal{T}(u)$ always exclude some class, and as such the total bound according to Theorems 2, 4 and 5 is lower than if we just used the naive bound. In the case of binary classification, for example, the bounds are only the same if there is some set of teachers that are *always* unanimous when classifying new samples.

**Remarks.** (i) $m(x_{new}; u)$ is smallest when the teachers in $\mathcal{T}(u)$ are divided evenly among the classes when classifying $x_{new}$, this is intuitive because in such a situation, $u$ is providing very little information about how to classify $u$ and thus little is being leaked about $u$ when we classify $x_{new}$.
(ii) $m(u)$ is bounded below by $1 - \frac{1}{|\mathcal{C}|}$ and so our method will provide the biggest improvements for binary classification and the improvements will decay as the number of classes increases.
(iii) When $k = 1$, $m(u)$ is always 1 because $n_c$ is 1 for some $c \in \mathcal{C}$ and then 0 for all remaining classes and from this we recover the standard bound of $2\lambda l(l+1)$ used for subsample-and-aggregate.
(iv) For Eq. 3 and 4 to be equal, there must exist some $u^*$ for which $m(x_{new}; u^*) > m(x_{new}; u)$ for *all* $u$ and $x_{new}$. This amounts to there being some set of teachers (corresponding to $u^*$) that are in more agreement than *every* other set of teachers for *every* new point they are asked to classify. Other than in this unlikely scenario, Eq. 4 will be *strictly* smaller than Eq. 3.

### 4.4 Semi-supervised knowledge transfer

We now discuss how best to leverage the fact that the best gains from our approach come from answering several queries (as implied by equations 3 and 4). We first note that the vanilla subsample-and-aggregate method does not derive data-dependent privacy guarantees for an individual query, and thus, for a fixed $\epsilon$ and $\delta$, the number of queries that can be answered by the mechanism is known in advance. In contrast, because our data-driven bounds on the personalised moments accountants

depend on the queries themselves, the cost of any given query is not known in advance and as such the number of queries we *can* answer before using up our privacy allowance ($\epsilon$) is unknown.

Unfortunately, we cannot simply answer queries until the allowance is used up, because the number of queries that we answer is a function of the data itself and thus we would need to introduce a differentially private mechanism for determining when to stop (such as calculating $\epsilon$ and $\delta$ after each query using smooth-sensitivity as proposed in [8]). Instead, we follow [8] and leverage the fact that we can answer more queries than standard subsample-and-aggregate to train a *student* model using unlabelled public data. The final output of our algorithm will then be a trained classifier that can be queried indefinitely. To train this model, we take unlabelled public data $\mathcal{P} = \{\tilde{x}_1, \tilde{x}_2, ...\}$ and label it using $\tilde{c}_\lambda$ until the privacy allowance has been used up. This will result in a (privately) labelled dataset $\tilde{\mathcal{P}} = \{(\tilde{x}_1, y_1), ..., (\tilde{x}_p, y_p)\}$ where $p$ is the number of queries answered. We train a student model, $S$, on this dataset and the resultant classifier can now be used to answer any future queries. Because of our data-driven bound on the personalised moments accountant, we will typically have that $p > q$ where $q$ is the number of queries that can be answered by a standard subsample-and-aggregate procedure. The pseudo-code for learning a differentially private student classifier using our differentially private bagging model is given in Algorithm 1 (pseudo-code for training a student model using standard subsample-and-aggregate is given in the Supplementary Materials for comparison). Note that the majority of for loops (in particular the one on line 18) can be parallelized.

---

**Algorithm 1** Semi-supervised differentially private knowledge transfer using multiple partitions

1: **Input:** $\epsilon, \delta, \mathcal{D}$, batch size $n_{mb}$, number of partitions $k$, number of teachers per partition $n$, noise size $\lambda$, maximum order of moments to be explored, $L$, unlabelled public data $\mathcal{D}_{pub}$
2: **Initialize:** $\{\theta_T^{i,j}\}_{i=1,j=1}^{k,n}, \theta_S, \hat{\epsilon} = 0, \alpha(l; x) = 0$ for $l = 1, ..., L, x \in \mathcal{D} \cup \{*\}$
3: Create $n$ partitions of the dataset which are each made up of $n$ disjoint subsets of the data $\mathcal{D}_{i,j}$, $i = 1, ..., n, j = 1, ..., k$ such that $\bigcup_i \mathcal{D}_{i,j} = \mathcal{D}$ and $\mathcal{D}_{i_1,j} \cap \mathcal{D}_{i_2,j} = \emptyset$ for all $i_1 \neq i_2, j$
4: Set $I(*) = \{(n,1), ..., (n,k)\}$
5: **while** Teachers have not converged **do**
6:     **for** $i = 1, ..., n$ **do**
7:         **for** $j = 1, ..., k$ **do**
8:             Sample $(\mathbf{x}_1, y_1), ..., (\mathbf{x}_{n_{mb}}, y_{n_{mb}}) \overset{\text{i.i.d.}}{\sim} D_{i,j}$
9:             Update teacher, $T_{i,j}$, using SGD
10:             $\nabla_{\theta_T^{i,j}} - \left[ \sum_{s=1}^{n_{mb}} \sum_{c \in \mathcal{C}} y_{s,c} \log(T_{i,j}^c(\mathbf{x}_s)) \right]$ (multi-task cross-entropy loss)
11: **while** $\hat{\epsilon} < \epsilon$ **do**
12:     Sample $\mathbf{x}_1, ..., \mathbf{x}_{n_{mb}} \sim \mathcal{D}_{pub}$
13:     **for** $s = 1, ..., n_{mb}$ **do**
14:         $r_s \leftarrow \tilde{c}_\lambda(\mathbf{x}_s)$
15:         Update the element-wise moments accountants
16:         $n_c \leftarrow \frac{|\{(i,j):T_{i,j}(\mathbf{x}_s)=c\}|}{k}$ for $c \in \mathcal{C}$
17:         **for** $x \in \mathcal{D} \cup \{*\}$ **do**
18:             $n_c(x) \leftarrow \frac{|\{(i,j) \in I(x):T_{i,j}(\mathbf{x}_s)=c\}|}{k}$ for $c \in \mathcal{C}, x \in \mathcal{D}$
19:             $m(x) \leftarrow \max_c\{1 - n_c(x)\}$
20:             **for** $l = 1, ..., L$ **do**
21:                 $\alpha(l; x) \leftarrow \alpha(l; x) + 2\lambda^2 m(x)^2 l(l+1)$
22:     Update the student, $S$, using SGD
23:     $\nabla_{\theta_S} - \sum_{s=1}^{n_{mb}} \sum_{c \in \mathcal{C}} r_{s,c} \log S^c(\mathbf{x}_s)$ (multi-task cross-entropy loss)
24:     $\hat{\epsilon} \leftarrow \min_l \left[ \max_x \frac{\alpha(l;x) + \log(\frac{1}{\delta})}{l} \right]$
25: **Output:** $S$

---

**Theorem 6.** The output of Algorithm 1 is $(\epsilon, \delta)$-differentially private with respect to $\mathcal{D}$.

*Proof.* This follows from Theorems 2, 3, 4 and 5. $\qquad\square$

# 5 Experiments

In this section we compare our method (DPBag) against the standard subsample-and-aggregate framework (SAA) to illustrate the improvements that can be achieved at a fundamental level by using our model. Additionally, we compare against our method *without* the improved privacy bound (DPBAG-) to quantify the improvements that are due to the bagging procedure and those that are due to our improved privacy bound. We perform the experiments on two real-world datasets: Heart Failure and UCI Adult (dataset description and results for UCI Adult can be found in the Supplementary Materials). Implementation of DPBag can be found at `https://bitbucket.org/mvdschaar/mlforhealthlabpub/src/master/alg/dpbag/`.

**Heart Failure dataset:** The Heart Failure dataset is a private dataset consisting of 24175 patients who have suffered heart failure. We set the label of each patient as 3-year all-cause mortality, excluding all patients who are censored before 3 years. The total number of features is 29 and the number of patients is 24175. Among 24175 patients, 10387 patients (43.0%) die within 3 years.

We randomly divide the data into 3 disjoint subsets: (1) a training set (33%), (2) public data (33%), (3) a testing set (33%). In the main paper, we use logistic regression for the teacher and student models in both algorithms; additional results for Gradient Boosting Method (GBM) can be found in the Supplementary Materials. We set $\delta = 10^{-5}$. We vary $\epsilon \in \{1, 3, 5\}$, $n \in \{50, 100, 250\}$ and $k \in \{10, 50, 100\}$. In all cases we set $\lambda = \frac{2}{n}$. To save space, we report DPBag results for $n \in \{100, 250\}, k \in \{50, 100\}$ and SAA results for $n = 250$ (the best performing) in the main manuscript, with full tables reported in the Supplementary Materials. Results reported are the mean of 10 runs of each experiment.

## 5.1 Results

In Table 1 we report the accuracy, AUROC and AUPRC of the 3 methods and we also report these for a non-privately trained baseline model (NPB), allowing us to quantify how much has been "lost due to privacy". In Table 2, we report the total number of queries that could be made to each differentially private classifier before the privacy budget was used up.

In Table 1 we see that DPBag outperforms standard SAA for all values of $\epsilon$ with Table 2 showing that our method allows for a significant increase in the number of public samples that can be labelled (almost 100% more for $\epsilon = 3$).

The optimal number of teachers, $n$, varies with $\epsilon$, for both DPBag and SAA. We see that for $\epsilon = 1$, $n = 250$ performs best, but as we increase $\epsilon$ the optimal number of teachers decreases. For small $\epsilon$ and small $n$, very few public samples can be labelled and so the student does not have enough data to learn from. On the other hand, for large $\epsilon$ and large $n$, the number of answered queries is much larger, to the point where now the limiting factor is not the *number* of labels but is instead the *quality* of the labels. Since we scale the noise to the number of teachers, the label quality improves with fewer teachers because each teacher is trained on a larger portion of the training data. This is reflected by both DPBag and SAA. In the SAA results, the performance does not saturate as quickly with respect to $\epsilon$ because the number of queries that $\epsilon$ corresponds to for SAA is smaller than for DPBag.

As expected, we see that DPBAG- sits between SAA and DPBAG, enjoying performance gains due to a stronger underlying model, and thus more accurately labelled training samples for the student, but the improved privacy bound that DPBAG allows more samples to be labelled and thus further gains are still made.

Table 2 also sheds light on the behavior of DPBag with repsect to $k$. We see in Table 1 that both $k = 50$ and $k = 100$ can provide the best performance (depending on $n$ and $\epsilon$). In Table 2, the number of queries that can be answered increases with $k$. This implies that (as expected), as we increase $k$, the quantity $m(u)$ gets closer to 0.5, and so each query costs less. However, when $m(u)$ is close to 0.5 for *all* samples, $u$, in the dataset then neither class will have a clear majority and thus the labels are more susceptible to flipping due to the noise added. $k = 50$ appears to balance this trade-off when $\epsilon$ is larger (and so we can already answer more queries) and when $\epsilon$ is smaller we see that answering more queries is more important than answering them well, so $k = 100$ is preferred.

Table 1: Prediction performance (Accuracy, AUROC, AUPRC) of DPBag and SAA with $\delta = 10^{-5}$ on the Heart Failure dataset using Logistic Regression. **Bold** indicates the best performance achieved for the given metric and fixed $\epsilon$. DPBAG- is our method without the improved privacy analysis. NPB is a non-private baseline model, included to indicate an upper bound on our performance.

| Model | n | k | Accuracy | | | AUROC | | | AUPRC | | |
|---|---|---|---|---|---|---|---|---|---|---|---|
| | | | $\epsilon = 1$ | $\epsilon = 3$ | $\epsilon = 5$ | $\epsilon = 1$ | $\epsilon = 3$ | $\epsilon = 5$ | $\epsilon = 1$ | $\epsilon = 3$ | $\epsilon = 5$ |
| DPBag | 100 | 50 | .5639 | **.6085** | **.6154** | .5547 | .6326 | **.6453** | .4793 | .5530 | **.5656** |
| | | 100 | .5667 | .6050 | .6142 | .5626 | .6295 | .6448 | .4895 | .5496 | .5652 |
| | 250 | 50 | .5888 | .6061 | .6099 | .5954 | .6320 | .6391 | .5161 | .5526 | .5607 |
| | | 100 | **.5986** | .6077 | .6091 | **.6096** | **.6373** | .6398 | **.5289** | **.5542** | .5644 |
| DPBag- | 100 | 50 | .5614 | .6019 | .6128 | .5544 | .6288 | .6429 | .4792 | .5411 | .5612 |
| | | 100 | .5596 | .6007 | .6108 | .5609 | .6174 | .6354 | .4767 | .5338 | .5525 |
| | 250 | 50 | .5855 | .6051 | .6086 | .5896 | .6295 | .6366 | .5093 | .5498 | .5565 |
| | | 100 | .5875 | .6061 | .6110 | .5884 | .6321 | .6407 | .5103 | .5518 | .5615 |
| SAA | 250 | | .5798 | .6019 | .6024 | .5778 | .6284 | .6356 | .5023 | .5496 | .5559 |
| NPB | 1 | - | .6527 | | | .6992 | | | .6281 | | |

Table 2: Number of labels provided by each method before the privacy budget, $\epsilon$, is used up on the Heart Failure dataset. Note that DPBAG- and SAA have the same, non-data dependent privacy analysis and so provide the same number of labels as each other.

| Models | $n$ | $k$ | $\epsilon = 1$ | $\epsilon = 3$ | $\epsilon = 5$ |
|---|---|---|---|---|---|
| DPBag | 100 | 50 | 74 | 593 | 1487 |
| | | 100 | 76 | 609 | 1538 |
| | 250 | 50 | 468 | 3785 | 6380 |
| | | 100 | 507 | 4044 | 6805 |
| SAA | 250 | - | 264 | 2108 | 5269 |

# 6 Discussion

In this work, we introduced a new methodology for developing a differentially private classifier. Building on the ideas of subsample-and-aggregate, we divide the dataset *several* times, allowing us to derive (tighter) data-dependent bounds on the privacy cost of a query to our mechanism. To do so, we defined the personalised moments accountants, which we use to accumulate the privacy loss of a query with respect to each sample in the dataset (and any potentially added sample) individually.

A key advantage of our model, like subsample-and-aggregate is that it is model agnostic, and can be applied using any base learner, with the differential privacy guarantees holding regardless of the learner used.

We believe this work opens up several interesting avenues for future research: (i) the privacy guarantees could potentially be improved on by making assumptions about the base learners used, (ii) the personalised moments accountants naturally allow for the development of an algorithm that affords each sample a different level of differential privacy, i.e. personalised differential privacy [7], (iii) we believe bounds such as those derived in [8] and [9] that rely on the subsample-and-aggregate method will have natural analogs with respect to our bagging procedure corresponding to tighter bounds on the personalised moments accountants than can be shown for the global moments accountant using simple subsample-and-aggregate (see the discussion in the Supplementary Materials).

## Acknowledgments

This work was supported by the National Science Foundation (NSF grants 1462245 and 1533983), and the US Office of Naval Research (ONR).

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
