[Supplementary Material · DPBag_supp_materials_camera_ready_js_final.pdf]

# Supplementary Materials
# Differentially Private Bagging: Improved utility and cheaper privacy than subsample-and-aggregate

**James Jordon**
University of Oxford
james.jordon@wolfson.ox.ac.uk

**Jinsung Yoon**
University of California, Los Angeles
jsyoon0823@g.ucla.edu

**Mihaela van der Schaar**
University of Cambridge
University of California, Los Angeles
Alan Turing Institute
mv472@cam.ac.uk, mihaela@ee.ucla.edu

## 1 Discussion continued

As alluded to in the main paper, we believe that our work is a fundamental improvement over the subsample-and-aggregate model and as such can be used to improve (most) methods that build on the subsample-and-aggregate framework. To illustrate how we think this might look, we take Private Aggregation of Teacher Ensembles (PATE) [1] as an example (note that we do not prove the following result - it is a conjecture).

PATE builds on the standard subsample-and-aggregate model by deriving a data-dependent bound on the (global) moments accountant at each step of the form

$$\alpha(l) \leq \log((1-q)\left(\frac{1-q}{1-e^{2\lambda}q}\right)^l + qe^{2\lambda l}) \tag{1}$$

where

$$q = \sum_{c \neq c^*} \frac{2 + \lambda|n_c^* - n_c|}{4\exp(\lambda|n_c^* - n_c|)} \tag{2}$$

with $n_c^*$ denoting the number of teachers voting for the majority class.

What we believe our algorithm offers here is to improve on this data-dependent bound in a similar fashion to the improvement we make in the paper, which amounts to replacing $\lambda$ with $\lambda \times m$ and instead bounding the *personalised* moments accountants, so that

$$\check{\alpha}_{\tilde{c}_\lambda(x_{new})}(l; \mathcal{D}, u) \leq \log((1-q)\left(\frac{1-q}{1-e^{2\lambda m(x_{new};u)}q}\right)^l + qe^{2\lambda m(x_{new};u)l}). \tag{3}$$

While we are confident in this bound, it is unclear whether we can also replace the $\lambda$ term in the definition of $q$, which we leave as an open question for future research.

## 2 Full tables for the results on the Heart Failure dataset

### 2.1 Logistic Regression

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

Table 4: Number of labels provided by each method using GBM before the privacy budget, $\epsilon$, is used up on the Heart Failure dataset with $\delta = 10^{-5}$. Note that DPBAG- and SAA have the same, data-independent privacy analysis and so provide the same number of labels as each other.

| Models | $n$ | $k$ | $\epsilon = 1$ | $\epsilon = 3$ | $\epsilon = 5$ |
|---|---|---|---|---|---|
| DPBag | 100 | 50 | 73 | 615 | 1527 |
| | | 100 | 79 | 635 | 1591 |
| | 150 | 50 | 161 | 1332 | 3329 |
| | | 100 | 174 | 1386 | 3473 |
| SAA | 100 | - | 43 | 338 | 843 |
| | 150 | | 95 | 759 | 1897 |

# 3 Results on UCI Adult Dataset

**UCI Adult dataset:** UCI Adult dataset `https://archive.ics.uci.edu/ml/datasets/adult` is a public dataset for binary classification. The total number of features is 108 (after one-hot encoding) and the number of samples is 48841. Among 48841 samples, 11687 samples (23.9%) have class 1.

## 3.1 Logistic Regression

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

Table 8: Number of labels provided by each method using GBM before the privacy budget, $\epsilon$, is used up on the UCI Adult Dataset with $\delta = 10^{-5}$. Note that DPBAG- and SAA have the same, data-independent privacy analysis and so provide the same number of labels as each other.

| Models | $n$ | $k$ | $\epsilon = 1$ | $\epsilon = 3$ | $\epsilon = 5$ |
|---|---|---|---|---|---|
| DPBag | 100 | 50 | 51 | 413 | 1037 |
| | | 100 | 52 | 413 | 1042 |
| | 150 | 50 | 115 | 931 | 2323 |
| | | 100 | 117 | 933 | 2336 |
| SAA | 100 | - | 44 | 338 | 843 |
| | 150 | | 95 | 759 | 1897 |

# 4 Pseudo-code for Subsample-and-aggregate

---

**Algorithm 1** Semi-supervised differentially private knowledge transfer using subsample-and-aggregate

---

1: **Input:** $\epsilon$, $\delta$, $\mathcal{D}$, batch size $n_{mb}$, number of teachers $n$, noise size $\lambda$, maximum order of moments to be explored, $L$, unlabelled public data $\mathcal{D}_{pub}$

2: **Initialize:** $\{\theta_T^i\}_{i=1}^n$, $\theta_S$, $\hat{\epsilon} = 0$, $\alpha(l) = 0$ for $l = 1, ..., L$

3: Partition the dataset $n$ disjoint subsets $\mathcal{D}_i$, $i = 1, ..., n$ such that $\bigcup_i \mathcal{D}_i = \mathcal{D}$ and $\mathcal{D}_i \cap \mathcal{D}_j = \emptyset$ for all $i, j$

4: **while** Teachers have not converged **do**

5:     **for** $i = 1, ..., n$ **do**

6:         Sample $(\mathbf{x}_1, y_1), ..., (\mathbf{x}_{n_{mb}}, y_{n_{mb}}) \overset{\text{i.i.d.}}{\sim} D_i$

7:         Update teacher, $T_i$, using SGD

8:         $\nabla_{\theta_T^i} - \left[ \sum_{s=1}^{n_m b} \sum_{c \in \mathcal{C}} y_{s,c} \log(T_{i,j}^c(\mathbf{x}_s)) \right]$ (multi-task cross-entropy loss)

9: **while** $\hat{\epsilon} < \epsilon$ **do**

10:     Sample $\mathbf{x}_1, ..., \mathbf{x}_{n_{mb}} \sim \mathcal{D}_{pub}$

11:     **for** $s = 1, ..., n_{mb}$ **do**

12:         **for** $c \in \mathcal{C}$ **do**

13:             $n_c \leftarrow |\{(i,j) : T_{i,j}(\mathbf{x}_s) = c\}|$

14:         $r_s \leftarrow \arg\max\{n_c + Y_c : c \in \mathcal{C}\}$ where $Y_c$ are i.i.d. $Lap(\frac{1}{\lambda})$

15:         Update the moments accountants

16:         **for** $l = 1, ..., L$ **do**

17:             $\alpha(l) \leftarrow \alpha(l) + 2\lambda^2 l(l+1)$

18:     Update the student, $S$, using SGD

19:     $\nabla_{\theta_S} - \sum_{s=1}^{n_{mb}} \sum_{c \in \mathcal{C}} r_{s,c} \log S^c(\mathbf{x}_s)$ (multi-task cross-entropy loss)

20:     $\hat{\epsilon} \leftarrow \min_l \frac{\alpha(l) + \log(\frac{1}{\delta})}{l}$

21: **Output:** $S$

---

# References

[1] Nicolas Papernot, Martín Abadi, Ulfar Erlingsson, Ian Goodfellow, and Kunal Talwar. Semi-supervised knowledge transfer for deep learning from private training data. *arXiv preprint arXiv:1610.05755*, 2016.