[Reviews · NeurIPS 2019]

Reviewer 1



The authors consider differentially private “bagging.” A single dataset is divided into k partitions. Each partition splits the data in n subsets, for a total of kn subsets. This is analogous to classical bagging since a single datapoint can appear in multiple subsets (in this case k). Furthermore, each subset will be used to train a classifier. A final classifier takes the majority vote of the individual classifiers with Lap(k/lambda) noise added to n_c, the number of sub-classifiers that output label c. This immediately yields a lambda-DP bound on the counts, and therefore the label of the final classifier. The authors are next interested in answering a sequence of queries using this classifier. The key analytic tool for understand this setting is the personalized moments accountant, which allows for a stronger composition of the privacy guarantees for each individual query than simple composition, advanced composition, or even the standard moments accountant. Ordinarily the l-th moment accountant is defined with respect to the privacy-loss random variable defined on two neighboring datasets. The privacy random variable is defined by drawing an outcome from the mechanism applied first dataset and computing the log odds ratio of the outcome with respect to the neighboring dataset. The worst-case (across datasets) l-th moment of this random variable essentially defines the lth moments accountant. In this work, the authors split the definition above into neighboring datasets that add an element, and neighboring datasets that subtract an element. They then show that all the same composition theorems hold for these “upwards” and “downwards” moments accountants individually. Much of the proofs here simply mirror the original moments accountant. They offer this as a tool for attaining better privacy bounds by first applying composition before taking the maximum between the upwards and downwards accountants. With this decomposition they can bound the moments accountant for the bagging setting by a data-dependent quantity. Given a new query point, for each class c and user u, we ask what fraction of classifiers that utilize user u’s data classify this point as c. If there is any user and class for which this fraction is 1, the personalized moments accountant yields a bound equivalent to the standard moments accountant. However, if the query point induces disagreement for all users, the bound is strictly better than the moments accountant. Across many query points, we should expect the latter case to sometimes happen, allowing us to use less privacy budget (although any budgeting will be data-dependent). This is born out in the experiments provided in the paper. The paper is well-written and easy to understand.

Reviewer 2



The authors modify the moments accountant analysis of Abadi et al for a subsample-and-aggregate style algorithm. The key technical idea is that their analysis treats differently privacy loss when a data point is removed vs added. The results seem plausible and rigorous (although I did not verify all details), but I wish more effort had gone toward comparing the results here to the analog without the separate downwards and upwards moment accounting to help show the power of this technique. At many times the prose was too high-level/imprecise to help me understand the significance of the pieces not immediately inherited from Abadi et al. Comments: *Avoid opinion language like “we believe” in comparing techniques qualitatively and speculating about their future impact. *The paragraph before 4.2 seems to be the main idea, but it could use some clarification. How much better should we expect (4) to be than (3)? You make a comment about how it is “unlikely” that the two bounds are the same, but what does unlikely mean in this sentence? More rigorous claims along these lines could strengthen the paper. *The paper should be reorganized so Section 4 is (at least almost) all new contributions; as it is, almost all of 4.1 is just inherited from Abadi et al. *Use $\ell$ instead of $l$ for readability. *Is there a typo in Thm 2? alpha does not appear to be defined with u as a parameter. *Thm 3: “The mechanism” -> “Any mechanism” *m is defined to be the the fraction of teachers that voted for all but the least popular vote c_min, which is different from the claim at Line 203 that unanimity is the only way to get m=1. Thus Line 203 seems to be an overstatement. Can you clarify? *The simulations are useful baselines, but a formal accuracy guarantee is really required in 4.3 to assess the quality of this technique.

Reviewer 3



The paper explores a remarkably simple but consequential improvement on the standard sample-and-aggregate framework. We have two main concerns about the paper. First, it is the relatively niche appeal of the result - for pragmatical reasons, sample-and-aggregate, or PATE, frameworks are very rarely used. Second, the paper compares its personalized accountant mechanism with the original sample-and-aggregate, outperforming it slightly. A more relevant comparison would have been with either PATE, or the "Scalable PATE" (ICLR 2018), both of which apply their own versions of data-dependent accounting mechanisms. In its current form the paper is somewhat half-baked: rather than improving on the latest state-of-the-art, it uses as the main benchmark a 2007 paper.

[Author Response · NeurIPS 2019]

We thank all the reviewers for their insightful comments.

**Reviewer 2**: **Answer 1** - We have conducted a further experiment in which we use the bagging classifier but the simple, non-data-dependent bound on the privacy (DPBAG-) to demonstrate how much is gained from the bagging classifier and how much is gained from the improved privacy bound we derive. The table below shows the results for each $\epsilon$ and each metric for the best choice of $n$ and $k$ (which are not the same across DPBAG and DPBAG-). The full table will be included in the revised manuscript. (Also see **Reviewer 3: Answer 3** for results with GBM as the base learner.)

| Model | Accuracy | | | AUROC | | | AUPRC | | |
|---|---|---|---|---|---|---|---|---|---|
| | $\epsilon = 1$ | $\epsilon = 3$ | $\epsilon = 5$ | $\epsilon = 1$ | $\epsilon = 3$ | $\epsilon = 5$ | $\epsilon = 1$ | $\epsilon = 3$ | $\epsilon = 5$ |
| DPBag | .5986 | .6085 | .6154 | .6096 | .6373 | .6453 | .5289 | .5542 | .5656 |
| DPBag- | .5875 | .6061 | .6128 | .5896 | .6321 | .6429 | .5103 | .5518 | .5615 |

**Answer 2** - We will revise the related works section, removing any opinion language, in the revised manuscript.

**Answer 3** - (3) and (4) will only be equal when there is some personalised moments accountant that dominates all other personalised moments accountants for *every* query. If we consider the bounds derived in 4.2 for our personalised moments accountant, we see that this would mean that $m(x_{new}; u^*) > m(x_{new}, u)$ for some $u^*$, for **all** $u$ and for **all** $x_{new}$. This corresponds to there being some set of teachers (corresponding to $u^*$) which essentially always disagree on every queried $x_{new}$. We will clarify this in the revised manuscript.

**Answer 4** - While the personalised moments accountant is introduced as an intermediate quantity in Abadi et. al, our defining it as an explicit quantity which we then go on to show can lead to meaningful analysis justifies its inclusion in a section dedicated to our contributions. In order to separate Theorem's 2 and 3 which are inherited from Abadi et al. we will create a new subsection within section 4 that makes this clear. Thank you for the suggestion.

**Answer 5** - Thank you, we will replace $l$ with $\ell$ and "the mechanism" with "any mechanism".

**Answer 6** - Theorem 2 is being stated with respect to the personalised moments accountants, for which the downwards accountant does depend on $u$. There is however a typo on line 158, the RHS of the inequality should be $\check{\alpha}$ rather than $\alpha$. We will correct this in the revised manuscript.

**Answer 7** - Thank you, this is correct. $m = 1$ when there is any class for which no teachers vote, rather than there being unanimity. In the case of binary classification, these two are the same, which is where the confusing language originated. We will correct this in the main manuscript.

**Answer 8** - We agree that a theoretical result for accuracy would be nice, however, we have yet to derive one. We have since conducted a further experiment using GBM as the base learner (in place of logistic regression) to further verify that DPBag outperforms standard subsample-and-aggregate empirically, see Answer 3 to Reviewer 3.

**Reviewer 3**: **Answer 1** - While it is true that the main contribution of the paper is the bagging variation of subsample-and-aggregate, the explicit definition and demonstration of utility of the personalised moments accountants amount to a contribution that may have consequences for other techniques. In addition, the PATE framework was recently shown to be a useful and practical tool for building a differentially private GAN in (Jordon et al. 2019).

**Answer 2** - As stated in the paper, our work in this paper is in parallel to the work of PATE, with improvements made in this paper being applicable to PATE as well. The improvements are over the underlying subsample-and-aggregate framework rather than a different privacy analysis (which is what PATE provides). We conjecture that the privacy bounds in PATE can be translated over to our work in much the same way the naive bounds can be (see the Supplementary Materials, section 1 for details). Moreover, we have ran experiments with PATE and found that their data-dependent bound was very rarely, or never, smaller than the naive bound, and as such PATE simply reduced to standard subsample-and-aggregate. But we stress that the key reason for comparing with standard subsample-and-aggregate is because we believe both PATE and Scalable PATE can be applied **on top** of our method.

**Answer 3** - We have conducted additional experiments using gradient boosting method (GBM) as the base classifier for the teachers. The following table shows the results for DPBAG, DPBAG- (see **Reviewer 2: Answer 1**) and SAA (with the best setting of $n$ and $k$ for each metric and $\epsilon$). The full table will be included in the revised supplementary materials.

| Model | Accuracy | | | AUROC | | | AUPRC | | |
|---|---|---|---|---|---|---|---|---|---|
| | $\epsilon = 1$ | $\epsilon = 3$ | $\epsilon = 5$ | $\epsilon = 1$ | $\epsilon = 3$ | $\epsilon = 5$ | $\epsilon = 1$ | $\epsilon = 3$ | $\epsilon = 5$ |
| DPBag | .5912 | .6165 | .6239 | .5987 | .6289 | .6451 | .5182 | .5504 | .5691 |
| DPBAG- | .5786 | .6061 | .6186 | .5911 | .6203 | .6355 | .5158 | .5433 | .5556 |
| SAA | .5763 | .5977 | .6111 | .5839 | .6137 | .6276 | .5005 | .5353 | .5511 |

[Meta-Review · NeurIPS 2019]

Discussion and consideration of the author response led to one reviewer increasing their score significantly. In particular, the point that the improvements offered by the method here are actually complementary to those of PATE was convincing. The authors are encouraged to address the remaining reviewer comments in the final version of the paper. They should also, of course, make the improvements promised in the author response.